# SURPRISAL-DRIVEN FEEDBACK IN RECURRENT NETWORKS

**Kamil Rocki**
IBM Research
San Jose, CA 95120, USA
kmrocki@us.ibm.com

## ABSTRACT

Recurrent neural nets are widely used for predicting temporal data. Their inherent deep feedforward structure allows learning complex sequential patterns. It is believed that top-down feedback might be an important missing ingredient which in theory could help disambiguate similar patterns depending on broader context. In this paper, we introduce surprisal-driven recurrent networks, which take into account past error information when making new predictions. This is achieved by continuously monitoring the discrepancy between most recent predictions and the actual observations. Furthermore, we show that it outperforms other stochastic and fully deterministic approaches on enwik8 character level prediction task achieving 1.37 BPC.

## 1 INTRODUCTION

Based on human performance on the same task, it is believed that an important ingredient which is missing in state-of-the-art variants of recurrent networks is top-down feedback. Despite evidence of its existence, it is not entirely clear how mammalian brain might implement such a mechanism. It is important to understand what kind of top-down interaction contributes to improved prediction capability in order to tackle more challenging AI problems requiring interpretation of deeper contextual information. Furthermore, it might provide clues as what makes human cognitive abilities so unique. Existing approaches which consider top-down feedback in neural networks are primarily focused on stacked layers of neurons, where higher-level representations constitute a top-down signal source. In this paper, we propose that the discrepancy between most recent predictions and observations might be effectively used as a feedback signal affecting further predictions. It is very common to use such a discrepancy during learning phase as the error which is subject to minimization, but not during inference. We show that is also possible to use such top-down signal without losing generality of the algorithm and that it improves generalization capabilities when applied to Long-Short Term Memory (Hochreiter & Schmidhuber, 1997) architecture. It is important to point out that the feedback idea presented here applies only to temporal data.

### 1.1 SUMMARY OF CONTRIBUTIONS

The main contributions of this work are:

- the introduction of a novel way of incorporating most recent misprediction measure as an additional input signal
- extending state-of-the-art performance on character-level text modeling using Hutter Wikipedia dataset.

### 1.2 RELATED WORK

There exist other approaches which attempted to introduce top-down input for improving predictions. One such architecture is Gated-Feedback RNN (Chung et al., 2015). An important difference between architecture proposed here and theirs is the source of the feedback signal. In GF-RNN it is assumed that there exist higher level representation layers and they constitute the feedback source.

On the other hand, here, feedback depends directly on the discrepancy between past predictions and current observation and operates even within a single layer. Another related concept is Ladder Networks (Rasmus et al., 2015), where top-down connections contribute to improved semi-supervised learning performance.

## 2 FEEDBACK: MISPREDICTION-DRIVEN PREDICTION

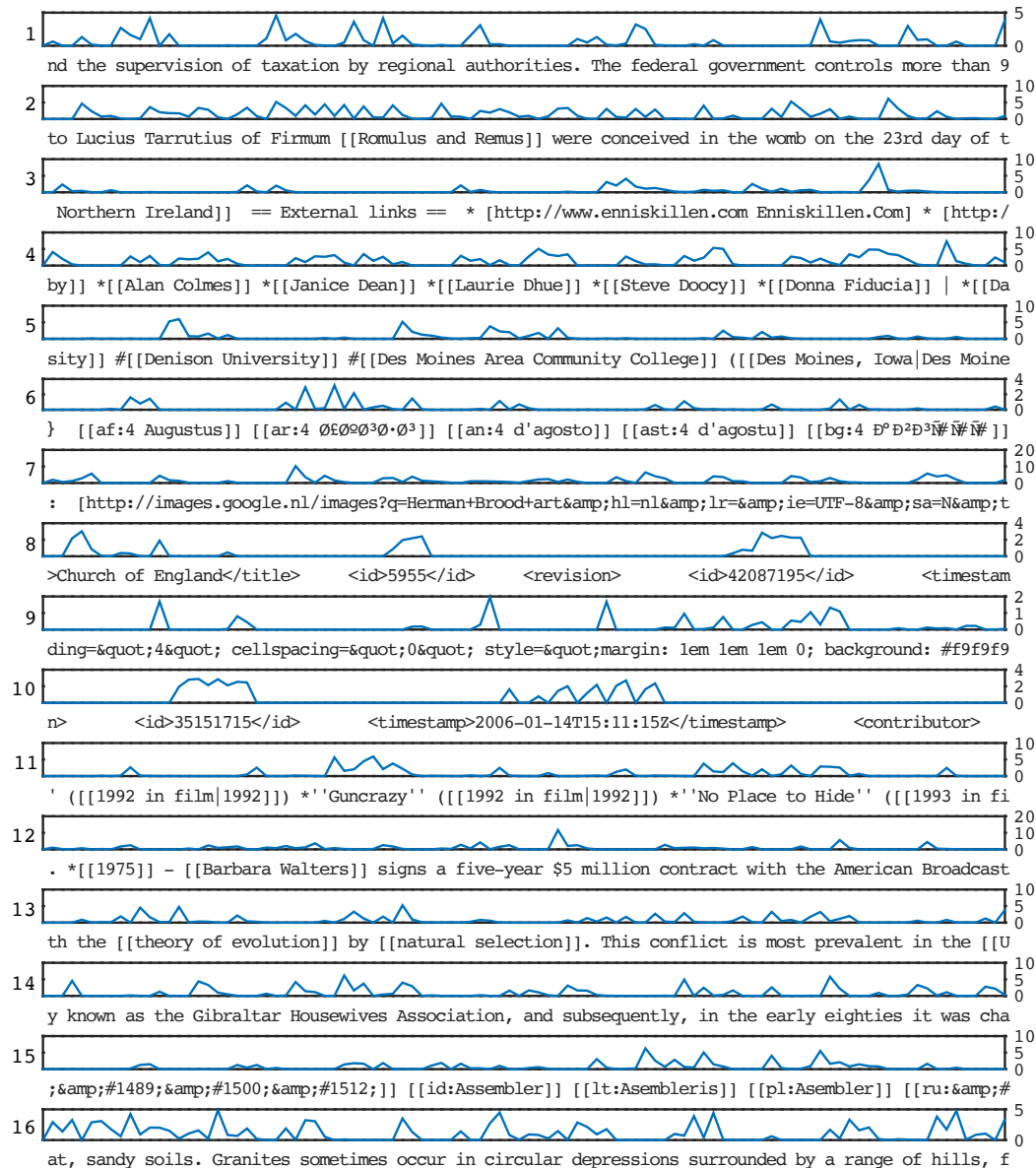

Figure 1: Illustration of $s_t$ signal on a typical batch of 16 sequences of length 100 from enwik8 dataset. $y$-axis is negative log probability in bits. Intuitively surprise signal is low when a text fragment is highly predictable (i.e. in the $< timestamp >$ part - sequence no 10, the tag itself is highly predictable, whereas the exact date cannot be predicted and should not be the focus of attention). The main idea presented in this paper is that feedback signal $s_t$ should be able to help in distinguishing predictable and inherently unpredictable parts during the inference phase.

## 2.1 NOTATION

The following notation is used throughout the section:

$x$ - inputs
$h$ - hidden units
$y$ - outputs
$p$ - output probabilities (normalized $y$)
$s$ - surprisal
$t$ - time step
$W$ - feedforward $x \rightarrow h$ connection matrix
$U$ - recurrent $h \rightarrow h$ connection matrix
$V$ - feedback $s \rightarrow h$ connection matrix
$S$ - truncated BPTT length
$M$ - number of inputs
$N$ - number of hidden units

$\cdot$ denotes matrix multiplication
$\odot$ denotes elementwise multiplication
$\sigma(\cdot), tanh(\cdot)$ - elementwise nonlinearities
$\delta x = \frac{\partial E}{\partial x}$

In case of LSTM, the following concatenated representations are used:

$$g_t = \begin{bmatrix} i_t \\ f_t \\ o_t \\ u_t \end{bmatrix} \quad b = \begin{bmatrix} b^i \\ b_f \\ b_o \\ b_u \end{bmatrix} \quad U = \begin{bmatrix} U^i \\ U_f \\ U_o \\ U_u \end{bmatrix} \quad W = \begin{bmatrix} W^i \\ W_f \\ W_o \\ W_u \end{bmatrix} \quad V = \begin{bmatrix} V^i \\ V_f \\ V_o \\ V_u \end{bmatrix} \tag{1}$$

## 2.2 SIMPLE RNN WITHOUT FEEDBACK

First, we show a simple recurrent neural network architecture without feedback which serves as a basis for demonstrating our approach. It is illustrated in Fig. 2 and formulated as follows:

$$h_t = tanh(W \cdot x_t + U \cdot h_{t-1} + b) \tag{2}$$

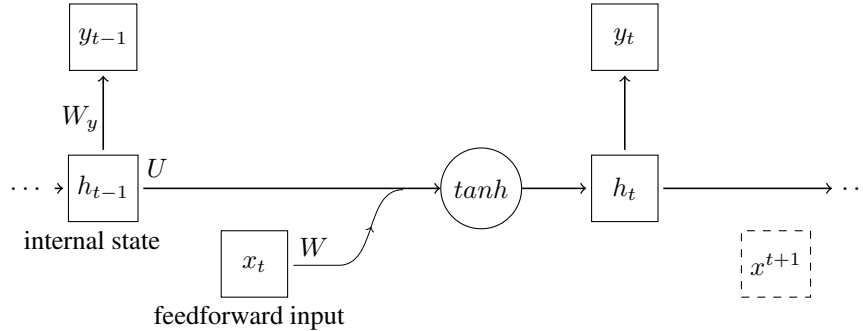

Figure 2: Simple RNN; $h$ - internal (hidden) states; $x$ are inputs, $y$ are optional outputs to be emitted

## 2.3 FEEDBACK AUGMENTED RECURRENT NETWORKS

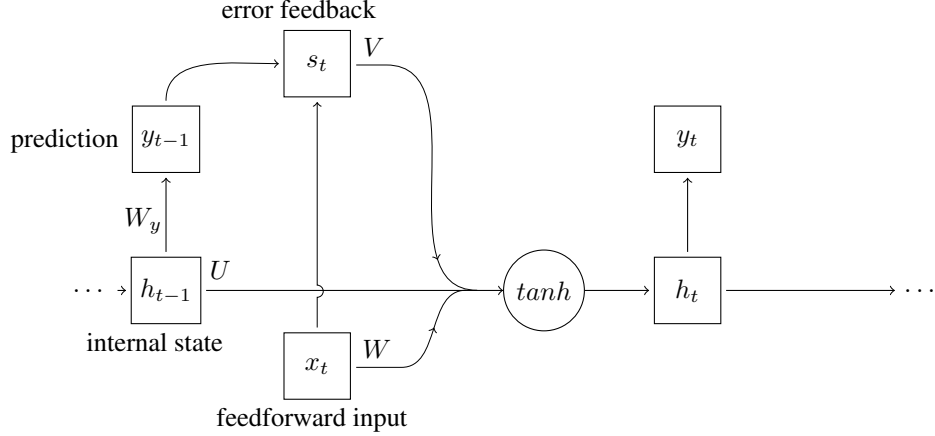

Figure 3: Surprisal-Feedback RNN; $s_t$ represents surprisal (in information theory sense) - the discrepancy between prediction at time step $t-1$ and the actual observation at time step $t$; it constitutes additional input signal to be considered when making a prediction for the next time step.

Figure 3 presents the main idea of surprisal-driven feedback in recurrent networks. In addition to feedforward and recurrent connections $W$ and $U$, we added one additional matrix $V$. One more input signal, namely $V \cdot s_t$ is being considered when updating hidden states of the network. We propose that the discrepancy $s_t$ between most recent predictions $p_{t-1}$ and observations $x_t$ might be effectively used as a feedback signal affecting further predictions. Such information is usually used during learning phase as an error signal, but not during inference. Our hypothesis is that it represents an important source of information which can be used during the inference phase, should be used and that it bring benefits in the form of improved generalization capability. Figure 1 presents examples of feedback signal being considered. Intuitively, when surprisal is near zero, the sum of input signals is the same as in a typical RNN. Next subsections provide mathematical description of the feedback architecture in terms of forward and backward passes for the Back Propagation Through Time (BPTT) (Werbos, 1990) algorithm.

## 2.4 FORWARD PASS

Set $h_0$, $c_0$ to zero and $p_0$ to uniform distribution or carry over the last state to emulate full BPTT.

$$\forall_i, p_0^i = \frac{1}{M}, i \in \{0, 1, .., M-1\}, t = 0 \tag{3}$$

for t = 1:1:S-1

I. Surprisal part

$$s_t = -\sum^i \log p_{t-1}^i \odot x_t^i \tag{4}$$

IIa. Computing hidden activities, Simple RNN

$$h_t = tanh(W \cdot x_t + U \cdot h_{t-1} + V \cdot s_t + b) \tag{5}$$

IIb. Computing hidden activities, LSTM (to be used instead of IIa)

$$f_t = \sigma(W_f \cdot x_t + U_f \cdot h_{t-1} + V_f \cdot s_t + b_f) \tag{6}$$

$$i_t = \sigma(W^i \cdot x_t + U^i \cdot h_{t-1} + V_i \cdot s_t + b_i) \tag{7}$$

$$o_t = \sigma(W_o \cdot x_t + U_o \cdot h_{t-1} + V_o \cdot s_t + b_o) \tag{8}$$

$$u_t = tanh(W_u \cdot x_t + U_u \cdot h_{t-1} + V_u \cdot s_t + b_u) \tag{9}$$

$$c_t = (1 - f_t) \odot c_{t-1} + i_t \odot u_t \tag{10}$$

$$\hat{c}_t = tanh(c_t) \tag{11}$$

$$h_t = o_t \odot \hat{c}_t \tag{12}$$

III. Outputs

$$y_t^i = W_y \cdot h_t + b_y \tag{13}$$

Softmax normalization

$$p_t^i = \frac{e^{y_t^i}}{\sum^i e^{y_t^i}} \tag{14}$$

## 2.5 BACKWARD PASS

```
for t = S-1:-1:1
```

I. Backprop through predictions

Backprop through softmax, cross-entropy error, accumulate

$$\frac{\partial E_t}{\partial y_t} = \frac{\partial E_t}{\partial y_t} + p_{t-1} - x_t \tag{15}$$

$\delta y \to \delta W_y, \delta b_y$

$$\frac{\partial E}{\partial W_y} = \frac{\partial E}{\partial W_y} + h_t^T \cdot \frac{\partial E_t}{\partial y_t} \tag{16}$$

$$\frac{\partial E}{\partial b_y} = \frac{\partial E}{\partial b_y} + \sum_{i=1}^{M} \frac{\partial E_t^i}{\partial y_t^i} \tag{17}$$

$\delta y \to \delta h$

$$\frac{\partial E_t}{\partial h_t} = \frac{\partial E_t}{\partial h_t} + \frac{\partial E_t}{\partial y_t} \cdot W_y^T \tag{18}$$

IIa. Backprop through hidden nonlinearity (simple RNN version)

$$\frac{\partial E_t}{\partial h_t} = \frac{\partial E_t}{\partial h_t} + \frac{\partial E_t}{\partial h_t} \odot tanh'(h_t) \tag{19}$$

$$\frac{\partial E_t}{\partial g_t} = \frac{\partial E_t}{\partial h_t} \tag{20}$$

IIb. Backprop through $c, h, g$ (LSTM version)

Backprop through memory cells, (keep gradients from the previous iteration)

$$\frac{\partial E_t}{\partial c_t} = \frac{\partial E_t}{\partial c_t} + \frac{\partial E_t}{\partial h_t} \odot o_t \odot tanh'(\hat{c}_t) \tag{21}$$

Carry error over to $\frac{\partial E_t}{\partial c_{t-1}}$

$$\frac{\partial E_t}{\partial c_{t-1}} = \frac{\partial E_t}{\partial c_{t-1}} + \frac{\partial E_t}{\partial c_t} \odot (1 - f_t) \tag{22}$$

Propagate error through the gates

$$\frac{\partial E_t}{\partial o_t} = \frac{\partial E_t}{\partial h_t} \odot \hat{c}_t \odot \sigma'(o_t) \tag{23}$$

$$\frac{\partial E_t}{\partial i_t} = \frac{\partial E_t}{\partial c_t} \odot u_t \odot \sigma'(i_t) \tag{24}$$

$$\frac{\partial E_t}{\partial f_t} = -\frac{\partial E_t}{\partial c_t} \odot c_{t-1} \odot \sigma'(f_t) \tag{25}$$

$$\frac{\partial E_t}{\partial u_t} = \frac{\partial E_t}{\partial c_t} \odot i_t \odot tanh'(u_t) \tag{26}$$

Carry error over to $\frac{\partial E_t}{\partial h_{t-1}}$

$$\frac{\partial E_t}{\partial h_{t-1}} = \frac{\partial E_t}{\partial g_t} \cdot U^T \tag{27}$$

III. Backprop through linearities

$$\frac{\partial E_t}{\partial b} = \frac{\partial E_t}{\partial b} + \sum_{i=1}^{N} \frac{\partial E_t}{\partial g_t^i} \tag{28}$$

$$\frac{\partial E}{\partial U} = \frac{\partial E}{\partial U} + h_{t-1}^T \cdot \frac{\partial E_t}{\partial g_t} \tag{29}$$

$$\frac{\partial E}{\partial W} = \frac{\partial E}{\partial W} + x_t^T \cdot \frac{\partial E_t}{\partial g_t} \tag{30}$$

$$\frac{\partial E}{\partial x} = \frac{\partial E}{\partial x} + \frac{\partial E_t}{\partial g_t} \cdot W^T \tag{31}$$

IV. Surprisal part

$$\frac{\partial E}{\partial V} = \frac{\partial E}{\partial V} + s_t^T \cdot \frac{\partial E_t}{\partial g_t} \tag{32}$$

$$\frac{\partial E}{\partial s_t} = \frac{\partial E}{\partial g_t} \cdot V^T \tag{33}$$

$$\frac{\partial E_t}{\partial p_{t-1}} = \frac{\partial E_t}{\partial s_t} \odot x_t \tag{34}$$

Adjust $\frac{\partial E_t}{\partial p_{t-1}}$ according to the sum of gradients and carry over to $\frac{\partial E_t}{\partial y_{t-1}}$

$$\frac{\partial E_t}{\partial y_{t-1}} = \frac{\partial E_t}{\partial p_{t-1}} - p_{t-1} \odot \sum_{i=1}^{M} \frac{\partial E_t}{\partial p_{t-1}^i} \tag{35}$$

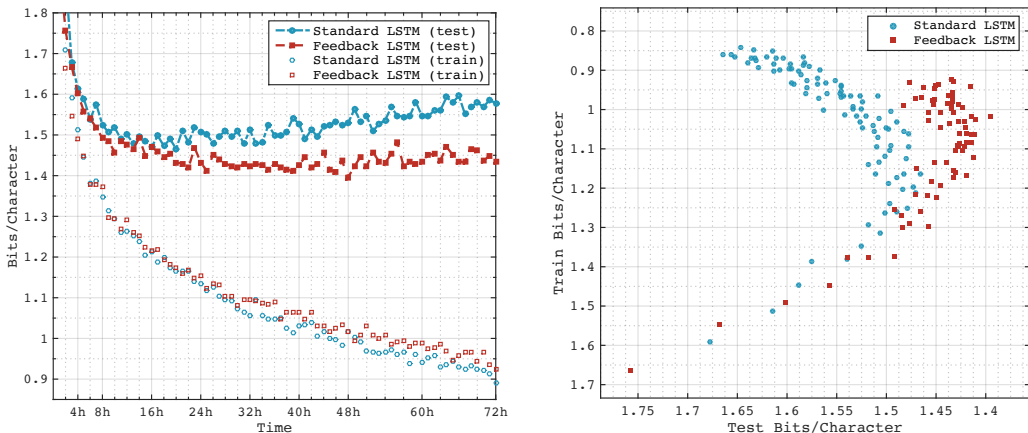

Figure 4: Training progress on enwik8 corpus, bits/character

## 3 EXPERIMENTS

We ran experiments on the enwik8 dataset. It constitutes first $10^8$ bytes of English Wikipedia dump (with all extra symbols present in XML), also known as Hutter Prize challenge dataset[2]. First 90% of each corpus was used for training, the next 5% for validation and the last 5% for reporting test accuracy. In each iteration sequences of length 10000 were randomly selected. The learning algorithm used was Adagrad[1] with a learning rate of 0.001. Weights were initialized using so-called Xavier initialization Glorot & Bengio (2010). Sequence length for BPTT was 100 and batch size 128, states were carried over for the entire sequence of 10000 emulating full BPTT. Forget bias was set initially to 1. Other parameters set to zero. The algorithm was written in C++ and CUDA 8 and ran on GTX Titan GPU for up to 10 days. Table 1 presents results comparing existing state-of-the-art approaches to the introduced Feedback LSTM algorithm which outperforms all other methods despite not having any regularizer.

Table 1: Bits per character on the Hutter Wikipedia dataset (test data).

|  | BPC |
| --- | --- |
| mRNN(Sutskever et al., 2011) | 1.60 |
| GF-RNN (Chung et al., 2015) | 1.58 |
| Grid LSTM (Kalchbrenner et al., 2015) | 1.47 |
| Standard LSTM[4] | 1.45 |
| MI-LSTM (Wu et al., 2016) | 1.44 |
| Recurrent Highway Networks (Zilly et al., 2016) | 1.42 |
| Array LSTM (Rocki, 2016) | 1.40 |
| Feedback LSTM | 1.39 |
| Hypernetworks (Ha et al., 2016) | 1.38 |
| Feedback LSTM + Zoneout (Krueger et al., 2016) | **1.37** |

## 4 SUMMARY

We introduced feedback recurrent network architecture, which takes advantage of temporal nature of the data and monitors the discrepancy between predictions and observations. This prediction error

---

[1]with a modification taking into consideration only recent window of gradient updates

[2]http://mattmahoney.net/dc/text.html

[3]This method does not belong to the 'dynamic evaluation' group: 1. It never actually sees test data during training. 2. It does not adapt weights during testing

[4]our implementation

information, also known as surprisal, is used when making new guesses. We showed that combining commonly used feedforward, recurrent and such feedback signals improves generalization capabilities of Long-Short Term Memory network. It outperforms other stochastic and fully deterministic approaches on enwik8 character level prediction achieving 1.37 BPC.

## 5 FURTHER WORK

It is still an open question what the feedback should really constitute as well as how it should interact with lower-level neurons (additive, multiplicative or another type of connection). Further improvements may be possible with the addition of regularization. Another research direction is incorporating sparsity in order improve disentangling sources of variation in temporal data.

## ACKNOWLEDGEMENTS

This work has been supported in part by the Defense Advanced Research Projects Agency (DARPA).

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
