# Peer review of "Surprisal-Driven Feedback in Recurrent Networks"

_ICLR 2017 — rejected_

[Public Comment · Kamil M Rocki · 13 Dec 2016]
**Source Code (C++/CUDA) for reproducing the results**

Source Code (C++/CUDA) for reproducing the results

[Official Review · AnonReviewer1 · rating 3 · confidence 5 · 16 Dec 2016]
**Need some revisions**

This paper proposes to leverage "surprisal" as top-down signal in RNN. More specifically author uses the error corresponding to the previous prediction as an extra input at the current timestep in a LSTM.

The general idea of suprising-driven feedback is interesting for online prediction task. It is a simple enough idea that seems to bring some significant improvements. However, the paper in its current form has some important flaws.

- Overall, the paper writing could be improved. In particular, section 2.4 and 2.5 is composed mostly by the equations of the forward and backward propagation of feedback RNN and feedback LSTM. However, author provides no analysis along with those equations. It is therefore not clear what insight the author tries to express in those sections. In addition, feedback RNN is not evaluated in the experimental section, so it is not clear why feedback RNN is described.

- The experimental evaluation is limited. Only one dataset enwik8 is explored. I think it is necessary to try the idea on different datasets to see if feedback LSTM sees some consistent improvements.
Also, author claims state-of-art on enwik8, but hypernetwork, already cited in the paper, achieves better results (1.34 BPC, table 4 in the hypernetworks paper).

- Author only compares to methods that do not use last prediction error as extra signal. I would argue that a comparison with dynamic evaluation would be more fair. 
 Feedback LSTM uses prediction error as extra input in the forward prop, while dynamic evaluation  backprop it through the network and change the weight accordingly. Also they don't propagate the prediction error in the same way, they both leverage "extra" supervised information through the prediction errors.


In summary:
Pros: 
- Interesting idea
- Seems to improve performances

Cons:
- Paper writing
- Weak evaluation (only one dataset)
- Compare only with approaches that does not use the last-timestep error signal

[Official Review · AnonReviewer3 · rating 3 · confidence 5 · 17 Dec 2016 (modified: 20 Dec 2016)]
**Misleading**

This paper proposes to use previous error signal of the output layer as an additional input to recurrent update function in order to enhance the modelling power of a dynamic system such as RNNs. 

-This paper makes an  erroneous assumption: test label information is not given in most of the real world applications, except few applications. This means that the language modelling task, which is the only experiment of this paper, may not be the right task to test this approach. Also, comparing against the models that do not use test error signal at inference time is unfair. We cannot just say that the test label information is being observed, this only holds in online-prediction problems.

-The experiment is only conducted on one dataset, reporting state-of-the-art result, but unfortunately this is not true. There are already more than four papers reporting better numbers than the one reported in this task, however the author did not cite them. I understand that this paper came before the other papers, but the manuscript should be updated before the final decision.

-The model size is still missing and without this information, it is hard to judge the contribution of the proposed trick.

[Official Review · AnonReviewer2 · rating 4 · confidence 4 · 20 Dec 2016]
**Badly Written**

Summary:
This paper proposes to use surprisal-driven feedback for training recurrent neural networks where they feedback the next-step prediction error of the network as an input to the network. Authors have shown a result on language modeling tasks.

Contributions:
The introduction of surprisal-driven feedback, which is just the feedback from the errors of the model from the previous time-steps.

Questions:
A point which is not fully clear from the paper is whether if you have used the ground-truth labels on the test set for the surprisal feedback part of the model? I assume that authors do that since they claim that they use the misprediction error as additional input.

Criticisms:
The paper is really badly written, authors should rethink the organization of the paper.
Most of the equations presented in the paper, about BPTT are not necessary for the main-text and could be moved to Appendix. 
The justification is not convincing enough.
Experimental results are lacking, only results on a single dataset are provided.
Although the authors claim that they got SOTA on enwiki8, there are other papers such as the HyperNetworks that got better results (1.34) than the result they achieve. This claim is wrong.
The model requires the ground-truth labels for the test-set, however, this assumption really limits the application of this technique to a very limited set of applications(more or less rules out most conditional language modeling tasks).

High-level Review:
    Pros: 
        - A simple modification of the model that seems to improve the results and it is an interesting modification.

    Cons:
       - The authors need to use test-set labels.
       - Writing of the paper is bad.
       - The authors assume that they have access to the ground-truth labels during the test-set.
       - Experimental results are lacking

[Final Decision · Program Chairs · 06 Feb 2017]
**ICLR committee final decision**

Based on the feedback, I'm going to be rejecting the paper on the following grounds:
 1. Results are not SOTA as reported.
 2. No real experiments other than cursory experiments on Hutter prize data.
 2. Writing is very poor.
 
 However, just for playing devil's advocate, to the reviewers, I would like to point out that I am in agreement with the author that dynamic evaluation is not equivalent to this method. The weights are not changed in this model, as far as I can see, for the test set. Surprisal is just an extra input to the model. I think the reviewers were puzzled by the fact that at test time, the actual sequence needs to be known. While this may be problematic for generative modeling, I do not see why this would be a problem for language modeling, where the goal of the model is only to provide a log prob to evaluate how good a sequence of text is. Long before language modeling started being used to generate text, this was the main reason to use it - in speech recognition, spelling correction etc..